# Nursing students' attitude on the practice of e-learning: A cross-sectional survey amid COVID-19 in Nepal

**Pratima Thapa** [1]*, **Suman Lata Bhandari** [2], **Sharada Pathak**[3]

**1** Department of Nursing, Maternal Health Nursing, College of Medical Sciences, Kathmandu University, Chitwan, Nepal, **2** Department of Nursing, Medical-Surgical Nursing, Manmohan Memorial Institute of Health Sciences, Tribhuvan University, Kathmandu, Nepal, **3** Department of Nursing, Medical-Surgical Nursing, Chitwan Medical College, Tribhuvan University, Chitwan, Nepal

* pratiimathapa@gmail.com

**Data Availability Statement:** The datasets used and analyzed during the current study are available in supporting files.

**Funding:** The authors received no specific funding for this work.

## Abstract

### Introduction

In present days, the use of information technology (IT) in education is unquestionable. The mounting advancement of IT has changed the scenario of education. With the emergence of the current COVID-19 situation, it has undoubtedly provided a solution to most of our educational needs when all educational institutions remained closed due to the pandemic. This study aims to identify the nursing students' attitude towards the practice of e-learning amidst COVID-19.

### Methods

A descriptive web-based cross-sectional study was conducted among nursing students with a sample size of 470. A self-administered validated questionnaire along with a standard tool to measure the attitude was used for data collection. Data were analyzed using SPSS.

### Results

The mean ± SD age of the respondents was 20.91± 1.55 years. The majority (76.4%) of the respondents used mobile for their study and 90.4% used Wi-Fi for the internet source. The main advantage of e-learning was stated as the ability to stay at home (72.1%) followed by the reduced cost of accommodation and transport (51.3%) whereas the internet problem (81.7%) was the major disadvantage followed by technical issues (65.5%). Only about 34% of the students found e-learning as effective as traditional face-to-face learning. The mean scores for the domains: perceived usefulness, intention to adapt, distant use of e-learning, ease of learning, technical support, and learning stressors were 3.1, 3.1, 3.8, 2.9, 2.9, and 2.5 respectively. Overall, 58.9% had a favorable attitude regarding e-learning. There was no significant association of overall attitude regarding e-learning with selected socio-demographic variables whereas it was positively associated with all of its six domains. All the domains were positively correlated with each other except for ease of learning with technical

**Competing interests:** The authors have declared that no competing interests exist.

**Abbreviations:** CMC, Chitwan Medical Sciences; CMS, College of Medical Sciences; COVID-19, Corona Virus Disease 2019; KU, Kathmandu University; PU, Pokhara University; PurU, Purbanchal University; SMTC, Shree Medical and Technical College; SPSS, Statistical Package for Social Sciences; TU, Tribhuvan University.

support and distant use, and technical support with learning stressor and distant use. Learning stressor versus distant use was negatively correlated with each other.

## Conclusion

Though e-learning was implemented as a substitute during the pandemic, almost half of the nursing students showed a positive attitude regarding e-learning. The majority of the students had internet problems and technological issues. If e-learning can be made user-friendly with reduced technical barriers supplemented with programs that can enhance practical learning abilities, e-learning can be the vital alternative teaching method and learning in the nursing field.

## Background

The COVID-19 pandemic significantly disrupted every aspect of human life including the educational system. It caused chaos compelling educational institutions to suspend their regular activities [1]. The closure of schools and universities affected more than 1.5 billion students and youths across the globe [2]. In Nepal, the pandemic resulted in the closure of all educational institutions for almost nine months [3]. Many schools and universities started switching from traditional classroom teaching to virtual education methods to cope with the educational loss due to lockdown. Tribhuvan University (TU), one of the renowned and most prominent universities of Nepal, officially authorized to start online classes along with a guideline and the Ministry of Education also appealed to stakeholders to start classes using alternative methods [4].

The significance and efficacy of the implementation of e-learning have been investigated by previous studies. They report several reasons for its overall acceptability including its ease of use, flexibility, and better control over the environment. However, regardless of its rewards, there are quite a few limitations in e-learning such as social isolation, lack of student-teacher interaction, technical and connectivity problems [5]. In a study conducted in Iraq, lack of technical support was identified as one of the barriers to e-learning [6].

As the schools and colleges were closed for an indefinite period, both educational institutions and students experimented with several ways to complete their prescribed syllabus within the specified time frame in their academic calendar. Although, these measures created a degree of inconvenience among the faculty members it also allowed them to search for alternative methods using virtual mediums. This helped the transformation of traditional classroom teaching within a short period. Most universities shifted to online mode using Google meet, Microsoft Teams, Zoom, or other online platforms [7].

Online classes were the ultimate method for imparting education to students in the aftermath of COVID-19 [8]. As a result of the nationwide lockdown teachers were compelled to run their classes online, primarily by using Zoom, Google meet and several other methods that involve the internet. This growing trend of virtual education in the Nepali education system has increased the familiarity of Nepali teachers, students, and parents with online classes. For the majority it was a new experience [9].

For both practical and theory-oriented lessons, virtual mediums were used across different education fields to ensure the continuity of classes. Likewise, online education prevailed even in the medical field where learning is traditionally hands-on. Although the concept of e-learning is well established in developed countries, it is still novel in developing countries like ours.

In Nepal's medical field, e-learning is a new approach to teach students. In nursing education where most of the teaching-learning is physical, the pandemic compelled the use of virtual classes to complete the syllabus on time. Nevertheless, this teaching method can be more difficult compared to classroom teaching for both the teachers and students, as it takes time to get accustomed to the new approaches.

In this regard, it is very crucial to assess the opinion and viewpoint of students on virtual approaches to teaching and learning. Previous studies have evaluated and identified students' perceptions and attitudes towards e-learning during the COVID-19 pandemic. Most of the studies are from an international context while those from Nepal investigate students of non-medical background. To our knowledge, the use of e-learning in nursing education is a new approach in Nepal. This study was conducted to assess the attitude of nursing students towards e-learning at a time when it was the only available option to continue learning. The study is even more relevant in Nepal because e-learning had never been practiced on such a large scale before the pandemic. The results are expected to provide fresh insights into a field that has traditionally relied on physical learning within real-life settings in labs and wards to impart hands-on knowledge and skills.

## Materials and methods

### Study design

Descriptive web-based cross-sectional study design was used. Four nursing colleges across four different universities were selected. The selected colleges were College of Medical Sciences- Kathmandu University (CMS-KU), Chitwan Medical College-Tribhuwan University (CMC-TU), Pokhara Nursing College-Pokhara University (PU), and Shree Medical and Technical College-Purbanchal University (SMTC-PurU) of Nepal. The total number of students studying nursing in these four colleges was 482. A total enumerative sampling method was used to determine the sample size. Data collection was performed online. The questionnaire was prepared in google forms and the link was shared in the Viber groups of the nursing students of all four colleges. Access to Viber groups was obtained from the administration section of each college. The total duration of the study was three months from August to October 2020.

### Ethical approval

Formal permission for ethical consideration was obtained from the Institutional Review Committee of College of Medical Sciences- Teaching Hospital (COMSTH-IRC), Bharatpur-10 (Ref no. 2020–079, NHRC ref no. 2586). Written permission from the administrative section of all the four colleges was obtained for data collection. Purpose and objectives of the study were clearly explained in the questionnaire form. Those who provided consent to participate in the study were asked to continue to fill the form. The form was prepared in such a way that whoever consented to participate in the study had to click on a proceed button for a response that they have gone through the consent form and agreed to participate in the study. Privacy and confidentiality of records was strictly maintained throughout the study.

### Study instrument

A Self-administered web-based questionnaire was developed through literature review. The questionnaire consisted of four sections where the first three parts were developed by the researchers themselves while the fourth part was adapted from previously published studies with authors' permission [10].

PART 1: Questionnaire related to socio-demographic information (age, college, year of study, gadgets used in e-learning, sources of the internet, and previous experience)

PART 2: Questionnaire related to advantages and disadvantages of e-learning (multiple choice questions were set and also they could write if they had any other felt advantages or disadvantages)

PART 3: Likert scale related to the effectiveness of e-learning against traditional face-to-face learning method (students had to compare e-learning with traditional learning using Likert scale 1 = strongly effective to 5 = strongly ineffective)

PART 4: Standard Likert scale measuring the attitude of students regarding e-learning. Questionnaire for Likert scale measuring attitude towards e-learning was adapted from a study conducted by Mehra V. et al. in India and Iran [11]. A modified shorter version of the tool was used in this study which was also used by Ali et.al. to measure the nursing students' attitude toward e-learning in Pakistan [10]. Necessary permission was obtained from the author of the study. The scale has six domains: Perceived usefulness (1 to 18), intention to adapt e-learning (19 to 27), ease of e-learning use (28 to 35), technical support (36 to 39), e-learning stressors (40 to 42), and distant use of e-learning (43 to 46). Score ranges from strongly disagree = 1 to strongly agree = 5. There are a total of 46 items with 26 positive items and 20 negative items. The total score ranges from 46 to 230. To assess the overall attitude, the mean score of the five-point Likert scale was considered. The reliability Cronbach's alpha score of the Likert scale for the domains: perceived usefulness, intention to adapt, ease, technical support, e-learning stressors, and distant use of e-learning was reported to be 0.75, 0.74, 0.70, 0.61, 0.79, and 0.71 respectively from a study conducted in Iran and India [10].

The validity of the instrument was ensured through extensive literature review and consultation with subject experts. The instrument was translated to Nepali language and then again translated to English version. The instrument's reliability was examined for internal consistency by pre-testing the instrument in 10% (48 nursing students) of a similar type of estimated population in a similar setting. The reliability score of the instrument for the part 2 and part 3 was found to be 0.98 on pre-testing.

## Statistical analysis

All collected data were checked, reviewed, coded, and organized for accuracy, completeness, and consistency. Out of 482, only 470 students participated in the study. Data were analyzed using the Statistical Package for Social Science (SPSS) version 16. Data were analyzed and interpreted using descriptive statistics (frequency, median, mean, percentage, and standard deviation) and inferential statistics. Chi-squared test was used to assess the relationship between attitude of the respondents towards e-learning and selected socio-demographic variables. Overall attitude was categorized as favorable and unfavorable from the mean score of the 5-point Likert scale. Pearson correlation was used to determine the relationship among the six domains.

## Results

### Socio-demographic data

Table 1 shows the socio-demographic data of the respondents. Of the total respondents, the majority (57.9%) were from the age group 20 to 25 years with a mean ± SD age of 20.91 ± 1.55 years. The majority (83.2%) of the respondents resided in urban areas. For most (68.9%) of the

**Table 1. Socio-demographic characteristics of the respondents (n = 470).**

| Variables | Frequency | Percentage |
|---|---|---|
| **Age of the respondents** *(Mean ± SD age: 20.91 ± 1.55)* | | |
| 16 to 20 | 195 | 41.5 |
| >20 to 25 | 272 | 57.9 |
| >25 to 30 | 3 | 0.6 |
| **Residence** | | |
| Rural | 79 | 16.8 |
| Urban | 391 | 83.2 |
| **Monthly family income (NRs)** | | |
| 15,000 to 50,000 | 324 | 68.9 |
| >50,000 to 1 Lakhs | 117 | 24.9 |
| >1 Lakhs | 29 | 6.2 |
| **Name of College** | | |
| Pokhara University, PU | 145 | 30.9 |
| College of Medical Sciences, KU | 126 | 26.8 |
| Chitwan Medical College, TU | 100 | 21.3 |
| Shree Medical College, Purbanchal University | 99 | 21.1 |
| **Year of Study** | | |
| 1st Year | 101 | 21.5 |
| 2nd Year | 124 | 26.4 |
| 3rd Year | 134 | 28.5 |
| 4th Year | 111 | 23.6 |
| **Gadgets Used** | | |
| Mobile | 359 | 76.4 |
| Computer | 6 | 1.3 |
| Laptop | 103 | 21.9 |
| Tablet | 2 | 0.4 |
| **Source of Internet** | | |
| Wi-Fi (Wireless Fidelity) | 425 | 90.4 |
| Telephone line | 22 | 4.7 |
| Mobile Data | 23 | 4.9 |
| **Ever participated in e-learning** | | |
| Yes | 42 | 8.9 |
| No | 428 | 91.1 |

respondents monthly family income varied between NRs. 15–50 thousand. It was found that 76.4% of the respondents used mobile phones for their e-learning and 90.4% used Wi-Fi as a source for internet. The majority (91.1%) of the participants had never participated in e-learning before this pandemic.

## Advantage and disadvantages of e-learning

Table 2 shows the advantages and disadvantages of e-learning. According to majority (72.1%) of the respondents the advantage of e-learning is the ability to stay at home followed by the reduced cost of accommodation and transport (51.3%), and the ability to record the meeting (38.1%). Regarding disadvantages of e-learning, majority (81.7%) of the respondents found it difficult due to internet problems followed by technical issues (65.5%) and reduced interaction with the patients (55.1%).

**Table 2. Advantages and disadvantages of e-learning (n = 470).**

| Advantages | Frequency | Percentage |
|---|---|---|
| Ability to stay at home | 339 | 72.1 |
| Classes interactivity | 68 | 14.5 |
| Ability to record meeting | 179 | 38.1 |
| Comfortable environment | 156 | 33.2 |
| Remote access | 42 | 8.9 |
| Reduce the cost of accommodation and transport | 241 | 51.3 |
| Learning in your environment | 1 | 0.2 |
| Time-saving | 1 | 0.2 |
| Helps to screenshot many beneficial slides | 1 | 0.2 |
| **Disadvantages** | | |
| Reduced interaction with patients | 259 | 55.1 |
| Poor learning condition at home | 185 | 39.4 |
| Lack of self-discipline | 112 | 23.8 |
| Reduced socialization | 103 | 21.9 |
| Internet problems | 384 | 81.7 |
| Technical issues | 308 | 65.5 |
| Poor interaction with facilitators | 213 | 45.3 |
| Eye problem | 1 | 0.2 |

Multiple response

Fig 1 illustrates the comparison between traditional learning and e-learning. E-learning was perceived to be less effective than traditional face-to-face learning with a mean score of 2.62.

## The attitude of respondents regarding e-learning

**Perceived usefulness.** Table 3 shows the perceived usefulness of e-learning among the respondents. With a mean score of 3.1, the majority (39.5%) of the respondents felt that e-learning can solve many of the educational problems. More than half (64.7%) said that it helps in saving their time and 40.8% expressed that e-learning improves their access to other learning material. However, most of the respondents felt that e-learning does not help achieve better result, increase learner's engagement in learning, improve teacher and student interaction, and improve understanding of the concepts (42.6%, 40.6%, 60.2%, and 38.3% respectively). Almost 40.0% felt that e-learning created problems rather than solve them, 41.3% believed that it is too time-consuming, 39.1% sensed that it had little impact on them and 31.5% said that it is not as informative as the teacher. Forty-three percent of the respondents agreed that e-learning can help replace other forms of teaching and learning, 38.9% said that it helped them in reinforcing their knowledge, 45.4% felt that it helped them organize their work, and half (50.4%) of them said that they could easily catch up on missed lectures. Furthermore, half (50.2%) of the respondents agreed that e-learning helps increase their effectiveness to create presentations and 58.9% think that it has increased their research capability. Similarly, 35.4% feel that universities should adapt e-learning for their students. The overall mean score for perceived usefulness was 3.1.

**Intention to adapt e-learning.** Table 4 represents the intention of the respondents to adapt to e-learning. The majority (44.5%) of the respondents disagreed on e-learning making them uncomfortable and 33.2% on calling it a medium for dehumanizing the process of

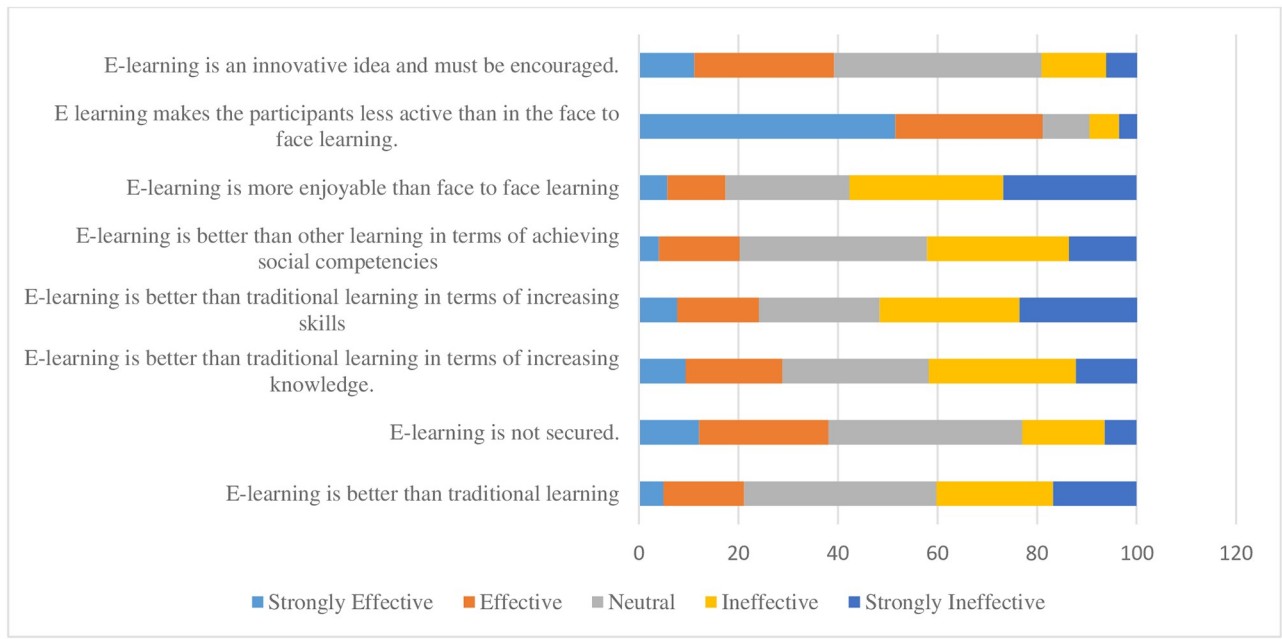

**Fig 1. Comparison of the effectiveness of e-learning against traditional face-to-face learning method (n = 470).**

**Table 3. Perceived usefulness (n = 470).**

| | Statements | SD | DA | N | A | SA | Mean ± SD |
|---|---|---|---|---|---|---|---|
| | **Perceived usefulness** | | | | | | 3.1 ± 0.5 |
| 1 | E-learning can solve many of the educational problems. | 31 (6.6) | 77 (16.4) | 176 (37.4) | 136 (28.9) | 50 (10.6) | 3.2± 1.1 |
| 2 | E-learning saves time. | 21 (4.5) | 38 (8.1) | 107 (22.8) | 192 (40.9) | 112 (23.8) | 3.7 ± 1.1 |
| 3 | E-learning improves access to learning material. | 36 (7.7) | 71 (15.1) | 172 (36.6) | 143 (30.4) | 48 (10.2) | 3.2 ± 1.1 |
| 4 | E-learning helps me to achieve better results. | 68 (14.5) | 132 (28.1) | 188 (40.0) | 67 (14.3) | 15 (3.2) | 2.6 ± 1.0 |
| 5 | E-learning increases learner's engagement in learning. | 47 (10.0) | 144 (30.6) | 165 (35.1) | 95 (20.2) | 19 (4.0) | 2.7 ± 1.0 |
| 6 | E- learning improves teacher and students interaction | 116 (24.7) | 167 (35.5.) | 135 (28.7) | 33 (7.0) | 19 (4.0) | 2.2 ± 1.0 |
| 7 | E-learning increases my understanding of concept | 55 (11.7) | 125 (26.6) | 199 (42.3) | 71 (15.1) | 20 (4.3) | 2.6 ± 0.9 |
| 8 | E-learning has created more problems than it solved. | 20 (4.3) | 102 (21.7) | 161 (34.3) | 118 (25.1) | 69 (14.7) | 2.8 ± 1.1 |
| 9 | E-learning is too time consuming to use. | 52 (11.1) | 142 (30.2) | 158 (33.6) | 75 (16.0) | 43 (9.1) | 3.2 ± 1.1 |
| 10 | E-learning has had little impact on me. | 13 (2.8) | 67 (14.3) | 206 (43.8) | 135 (28.7) | 49 (10.4) | 2.7 ± 0.9 |
| 11 | E-learning is as informative as the teacher. | 30 (6.4) | 118 (25.1) | 177 (37.7) | 111 (23.6) | 34 (7.2) | 3.0 ± 1.0 |
| 12 | E-learning will never replace other forms of teaching and learning. | 35 (7.4) | 92 (19.6) | 139 (29.6) | 126 (26.8) | 78 (16.6) | 2.7 ± 1.1 |
| 13 | E-learning help to reinforce my knowledge. | 17 (3.6) | 59 (12.6) | 211 (44.9) | 142 (30.2) | 41 (8.7) | 3.2 ± 0.9 |
| 14 | E-learning helps me to organize my work. | 9 (1.9) | 65 (13.8) | 183 (38.9) | 161 (34.3) | 52 (11.1) | 3.3 ± 0.9 |
| 15 | E-learning helps me to catch up missed lectures. | 44 (9.4) | 68 (14.5) | 121 (25.7) | 160 (34.0) | 77 (16.4) | 3.3 ± 1.1 |
| 16 | E-learning increases my effectiveness to create presentations. | 19(4.0) | 73 (15.5) | 142 (30.2) | 137 (29.1) | 99 (21.1) | 3.4 ± 1.1 |
| 17 | E-learning increases my research capability. | 17 (3.6) | 45 (9.6) | 131 (27.9) | 174 (37.0) | 103 (21.9) | 3.6 ± 1.0 |
| 18 | Universities should adapt e-learning for their students. | 53 (11.3) | 73 (15.5) | 178 (37.9) | 115 (24.5) | 51 (10.9) | 3.0 ± 1.1 |

SD: Strongly disagree, DA: Disagree, N: Neutral, A: Agree, SA: Strongly agree

**Table 4. Intention to adapt (n = 470).**

| | Statements | SD | DA | N | A | SA | Mean ± SD |
|---|---|---|---|---|---|---|---|
| | **Intention to adapt e-learning** | | | | | | 3.0 ± 0.6 |
| 1 | E-learning makes me uncomfortable because I don't understand it. | 55 (11.7) | 154 (32.8) | 145 (30.9) | 78 (16.6) | 38 (8.1) | 3.2 ± 1.1 |
| 2 | E-learning is a de-humanizing process of learning. | 49 (10.4) | 107(22.8) | 194 (41.3) | 85 (18.1) | 35 (7.4) | 3.1 ± 1.1 |
| 3 | I dislike the idea of using E-learning. | 80 (17.0) | 131 (27.9) | 149 (31.7) | 74 (15.7) | 36 (7.7) | 3.3 ± 1.1 |
| 4 | I am not in favor of E-learning as it leads to social isolation. | 46 (9.8) | 100 (21.3) | 147 (31.3) | 119 (25.3) | 58 (12.3) | 2.9 ± 1.1 |
| 5 | E-learning doesn't interest me. | 51 (10.9) | 136 (28.9) | 157 (33.4) | 87 (18.5) | 39 (8.3) | 3.1 ± 1.1 |
| 6 | I plan to participate in future e-learning courses. | 43 (9.1) | 88 (18.7) | 172 (36.6) | 119 (25.3) | 48 (10.2) | 3.1 ± 1.1 |
| 7 | I am planning to buy a computer to be able to follow lectures notes online. | 70 (14.9) | 104 (22.1) | 142 (30.2) | 97 (20.6) | 57 (12.1) | 2.9 ± 1.2 |
| 8 | Using E-learning makes learning fun. | 46 (9.8) | 81 (17.2) | 183 (38.9) | 124 (26.4) | 36 (7.7) | 3.1 ± 1.1 |
| 9 | I don't know what I would do without E-learning. | 53 (11.3) | 115 (24.5) | 168 (35.7) | 95 (20.2) | 39 (8.3) | 2.9 ± 1.1 |

SD: Strongly disagree, DA: Disagree, N: Neutral, A: Agree, SA: Strongly agree

learning. Almost 45% of the respondents differed in the notion of disliking the idea of using e-learning and 31.1% were in favor of e-learning as it is not causing so much social isolation. About 40% of the respondents were interested in using e-learning, 35% already planned to participate in future e-learning courses, 24.7% were planning to buy a computer. Learning was fun through e-learning for 34.1%. The overall mean score for the intention to adapt to e-learning was 3.1.

**Ease of learning.** Table 5 shows the ease in use of e-learning. For 45.5% of the respondents' use of e-learning was easier than using a library, 37.95% could easily use the web for lectures, 39.6% could learn the courses through the web, and acquiring any significant information from the internet was easy for 37.3% of the respondents. On the other hand, 47.2% felt that the use of the internet is making them slow and 54% said that technology can make them slaves sooner or later. The overall mean score for ease of e-learning use was 2.9.

**Technical support.** Table 6 represents the technical support provided by the respondents' institutions while e-learning was introduced and being practiced. The majority of the respondents had a neutral attitude towards technical support. Forty percent of the students had no idea regarding their institution's updated website and only 28.1% seek any assistance from college support services. About thirty percent of the students agreed on their institution providing

**Table 5. Ease of learning (n = 470).**

| | Statements | SD | DA | N | A | SA | Mean ± SD |
|---|---|---|---|---|---|---|---|
| | **Ease of learning** | | | | | | 2.9 ± 0.7 |
| 1 | Using E-learning is more difficult than using the library. | 82 (17.4) | 132 (28.1) | 136 (28.9) | 74 (15.7) | 46 (9.8) | 3.2 ± 1.2 |
| 2 | I can't read the lectures notes through the web. | 51 (10.9) | 127 (27) | 136 (28.9) | 86 (18.3) | 70 (14.9) | 3.0 ± 1.2 |
| 3 | I can't learn courses through the web. | 39 (8.3) | 147 (31.3) | 144 (30.6) | 92 (19.6) | 48 (10.2) | 3.1 ± 1.1 |
| 4 | It is difficult to acquire any significant information by using internet. | 51 (10.9) | 124 (26.4) | 137 (29.1) | 99 (21.1) | 59 (12.6) | 3.0 ± 1.1 |
| 5 | It is difficult to express my thoughts by writing through E- learning. | 23 (4.9) | 74 (15.7) | 148 (31.5) | 157 (33.4) | 68 (14.5) | 2.6 ± 1.1 |
| 6 | I find that using the internet make me slow. | 35 (7.4) | 88 (18.7) | 125 (26.6) | 126 (26.8) | 96 (20.4) | 2.6 ± 1.2 |
| 7 | I feel we are becoming slaves to technology. | 31 (6.6) | 48 (10.2) | 137 (29.1) | 136 (28.9) | 118 (25.1) | 2.4 ± 1.1 |
| 8 | My interaction with E-learning is not understandable. | 38 (8.1) | 134 (28.5) | 186 (39.6) | 77 (16.4) | 35 (7.4) | 3.1 ± 1.0 |

SD: Strongly disagree, DA: Disagree, N: Neutral, A: Agree, SA: Strongly agree

**Table 6. Technical support (n = 470).**

| | Statements | SD | DA | N | A | SA | Mean ± SD |
|---|---|---|---|---|---|---|---|
| | **Technical support** | | | | | | 2.9 ± 0.8 |
| 1 | My institute has an updated website. | 58 (12.3) | 85 (18.1) | 188 (40.0) | 107 (22.8) | 32 (6.8) | 2.9 ± 1.0 |
| 2 | My institute facilitates e-learning training program. | 56 (11.9) | 93 (19.8) | 182 (38.7) | 96 (20.4) | 43 (9.1) | 2.9 ± 1.1 |
| 3 | My institute has adequate technology for e-learning. | 66 (14.0) | 114 (24.3) | 166 (35.3) | 91 (19.4) | 33 (7.0) | 2.8 ± 1.1 |
| 4 | I seek technical assistance from college support services. | 56 (11.9) | 99 (21.1) | 183 (38.9) | 94 (20.0) | 38 (8.1) | 2.9 ± 1.0 |

SD: Strongly disagree, DA: Disagree, N: Neutral, A: Agree, SA: Strongly agree

**Table 7. Learning stressors (n = 470).**

| | Statements | SD | DA | N | A | SA | Mean ± SD |
|---|---|---|---|---|---|---|---|
| | **Learning stressor** | | | | | | 2.5 ± 0.6 |
| 1 | Feel anxious about my ability to use e learning effectively. | 39 (8.13) | 111 (23.6) | 177 (37.7) | 106 (22.6) | 37 (7.9) | 3.0 ± 1.0 |
| 2 | Slow internet connections stress me. | 9 (1.9) | 18 (3.8) | 41 (8.7) | 95 (20.2) | 307 (65.3) | 1.5 ± 0.9 |
| 3 | I feel pressured by my teachers to use E-learning for my research/ learning activities. | 67 (14.3) | 107 (22.8) | 133 (28.3) | 95 (20.2) | 68 (14.5) | 3.0 ± 1.2 |

SD: Strongly disagree, DA: Disagree, N: Neutral, A: Agree, SA: Strongly agree

e-learning training programs while 38.3% disagreed on their institution having adequate technology. The overall mean score for the provision of technical support for e-learning was 2.9.

**Learning stressors.** Table 7 shows the stressors that the respondents faced during e-learning. About thirty percent of the students felt anxious about their ability to use e-learning effectively. The majority 85.5% felt more stressed due to slow internet connection and 34.7% agreed on getting pressure from their teachers to use e-learning. The overall mean score for e-learning stressors was 2.5.

**Distant use of e-learning.** Table 8 represents the importance of distant use of e-learning. The majority (68.3%) supported the idea of e-learning as a medium to reach students living in remote areas and 66.7% agreed that it reduces travel related stress. Regarding e-learning to be adapted to allow married students to balance family and study demands and to allow working students to study from home, it was agreed upon by 55.8% and 67.7% of the respondents respectively. The overall mean score for distant use of e-learning was 3.8.

## The overall attitude of respondents regarding e-learning

Table 9 shows the overall attitude of the respondents regarding e-learning. Overall, 58.9% of the students had a positive attitude regarding e-learning.

**Table 8. Distant use of e-learning (n = 470).**

| | Statements | SD | DA | N | A | SA | Mean ± SD |
|---|---|---|---|---|---|---|---|
| | **Distant use of e-learning** | | | | | | 3.8 ± 0.7 |
| 1 | E-learning should be offered fully online to reach students living in remote areas. | 24 (5.1) | 29 (6.2) | 96 (20.4) | 134 (28.5) | 187 (39.8) | 3.9 ± 1.1 |
| 2 | E-learning should be used to reduce travel related stress. | 9 (1.9) | 24 (5.1) | 124 (26.4) | 177 (37.7) | 136 (28.9) | 3.8 ± 0.9 |
| 3 | E-learning should be adapted to allow married students to balance family and Study demands. | 26 (5.5) | 37 (7.9) | 145 (30.9) | 156 (33.2) | 106 (22.6) | 3.5 ± 1.0 |
| 4 | E-learning should be adapted to allow working students to study from home. | 10 (2.1) | 16 (3.4) | 126 (26.8) | 161 (34.3) | 157 (33.4) | 3.9 ± 0.9 |

SD: Strongly disagree, DA: Disagree, N: Neutral, A: Agree, SA: Strongly agree

**Table 9. Overall attitude of the respondents regarding e-learning (n = 470).**

| Characteristics | Response | | | |
|---|---|---|---|---|
| | Favorable ($\geq$ 60%) | | Unfavorable ($<$60%) | |
| | Frequency | Percent (%) | Frequency | Percent (%) |
| Overall Attitude | 277 | 58.9 | 193 | 41.1 |

Table 10 shows there was no statistically significant association between attitude regarding e-learning and selected socio-demographic variables like age, residence, college, year of study, and having participated in e-learning earlier.

Table 11 shows the domain wise response of students regarding e-learning with their respective p-values. All the domains except for intention to adapt were positively correlated with the students' overall attitude.

Table 12 shows the correlation between different domains of attitude regarding e-learning. All the domains were positively correlated with each other except for ease of learning with technical support and distant use, and technical support with learning stressor and distant use. Whereas learning stressor versus distant use was negatively correlated with one another.

## Discussion

The sudden closure of all the educational institutions due to the pandemic caused the educational sector to seek alternative practices to limit the interference on carrying out the regular teaching learning activities caused by the lockdown. The better way to deal with the situation came forward with the approach of practicing e-learning by academic institutions. The medical education sector also adapted similar approach, whereby the students were compelled to continue their education using e-learning approaches.

**Table 10. Association of attitude regarding e-learning with selected socio-demographic variables (n = 470).**

| Characteristics | Categories | Attitude | | | | p-value |
|---|---|---|---|---|---|---|
| | | Negative Attitude | | Positive Attitude | | |
| | | No. | % | No. | % | |
| Age in years | 16–20 | 84 | 43.1 | 111 | 56.9 | NA* |
| | 21–25 | 108 | 39.7 | 164 | 60.3 | |
| | 26–30 | 1 | 33.3 | 2 | 66.7 | |
| Residence | Rural | 34 | 43.0 | 45 | 57.0 | 0.71 |
| | Urban | 159 | 40.7 | 232 | 59.3 | |
| College | PU | 68 | 46.9 | 77 | 53.1 | 0.07 |
| | CMC, TU | 45 | 35.7 | 81 | 64.3 | |
| | CMS, KU | 34 | 34.0 | 66 | 66.0 | |
| | SMTC, PurU | 46 | 46.5 | 53 | 53.5 | |
| Year of study | 1st Year | 39 | 38.6 | 62 | 61.4 | 0.11 |
| | 2nd Year | 58 | 46.8 | 66 | 53.2 | |
| | 3rd Year | 60 | 44.8 | 74 | 55.2 | |
| | 4th Year | 36 | 32.4 | 75 | 67.6 | |
| Ever participated in e-learning before this pandemic | Yes | 14 | 50.0 | 28 | 66.7 | 0.28 |
| | No | 179 | 77.1 | 249 | 58.2 | |

*NA: Not applicable

**Table 11. Domain wise response of students regarding e-learning (n = 470).**

| Characteristics | Categories | Attitude | | | | p-value |
|---|---|---|---|---|---|---|
| | | Negative Attitude | | Positive Attitude | | |
| | | No. | % | No. | % | |
| Perceived usefulness | Negative | 161 | 84.7 | 29 | 15.3 | <0.001 |
| | Positive | 32 | 11.4 | 248 | 88.6 | |
| Intention to adapt | Negative | 0 | 0 | 0 | 0 | NA* |
| | Positive | 193 | 41.1 | 277 | 58.9 | |
| Ease of learning | Negative | 140 | 61.1 | 89 | 38.9 | <0.001 |
| | Positive | 53 | 22 | 188 | 78.0 | |
| Technical support | Negative | 103 | 50.2 | 102 | 49.8 | <0.001 |
| | Positive | 90 | 34.0 | 175 | 66.0 | |
| Learning stressor | Negative | 160 | 49.7 | 162 | 50.3 | <0.001 |
| | Positive | 33 | 22.3 | 115 | 77.7 | |
| Distant use | Negative | 30 | 78.9 | 8 | 21.1 | <0.001 |
| | Positive | 163 | 37.7 | 269 | 62.3 | |

*NA: Not applicable

This study explores nursing students' attitude regarding e-learning based on their experience with e-learning activities during the pandemic. This study was conducted among bachelor level nursing students, so it consisted only of female students with a mean age of 20.91 years which is similar to the study conducted in Indonesia among medical students [12].

Mobile phones, due to their flexibility and portability, became a popular e-learning gadget compared to laptops and computers during the COVID-19 pandemic. In this study it was

**Table 12. Correlation analysis of 6 domains of attitude regarding e-learning (n = 470).**

| Domains | r | p-value |
|---|---|---|
| Perceived usefulness vs intention to adapt | 0.685 | <0.01 |
| Perceived usefulness vs ease of learning | 0.466 | <0.01 |
| Perceived usefulness vs technical support | 0.217 | <0.01 |
| Perceived usefulness vs learning stressor | 0.200 | <0.01 |
| Perceived usefulness vs distance use | 0.305 | <0.01 |
| Intention to adapt vs ease of learning | 0.611 | <0.01 |
| Intention to adapt vs technical support | 0.143 | <0.01 |
| Intention to adapt vs learning stressor | 0.325 | <0.01 |
| Intention to adapt vs distant use | 0.262 | <0.01 |
| Ease of learning vs technical support | 0.020 | 0.659 |
| Ease of learning vs learning stressor | 0.452 | <0.01 |
| Ease of learning vs distant use | 0.005 | 0.91 |
| Technical support vs learning stressor | 0.005 | 0.91 |
| Technical support vs distant use | 0.087 | 0.59 |
| Learning stressor vs distant use | -0.093 | <0.05 |

**significant at 0.01

*significant at 0.05

r = value of Pearson correlation

found that majority of the respondents used mobile phones for their e-learning which is similar to the finding of a study conducted in Pakistan [5]. However, in one of the studies conducted in India the use of mobile phones and laptops was approximately fifty-fifty [13]. A study conducted in Australia found that mobiles are popular because learning can take place anytime and anywhere using them. Most of the respondents in the present study used Wi-Fi as a source of internet rather than cellular data and telephone data which is similar to the study conducted in Nepal among BDS (Bachelor of Dental Surgery) students [14]. In contrast to the finding of this study, one study reported that majority of the students participated in the study had used data packs for their online class [15]. The result of the study conducted among BDS students in Nepal was also similar to this study in the case of the percentage of respondents who had never participated in e-learning before this pandemic. This may be due to scarce practice of e-learning in Nepal before the COVID-19 pandemic. E-learning was only in its nascent stage before COVID-19 pandemic in Nepal.

## Advantage and disadvantages of e-learning

The ability to stay at home was the major advantage of e-learning for the students in this study which was similar to the study conducted in Poland [16]. Apart from this, the reduction in accommodation and transport costs was a key benefit of e-learning in this study which is similar to another study [17]. When courses are available entirely online, distance does not become a major factor in getting education because physical presence is not required in college [18].

Internet problems and technical issues were the major disadvantages of e-learning that the students experienced the most. These findings are parallel to the findings of other studies [6, 15, 19]. Lack of IT (Information Technology) skills and reluctance to use e-learning among the academics and students were cited as the major barriers in implementing e-learning in a study conducted in UK [20]. Furthermore, lack of interaction with the patients was also stated as a disadvantage by almost half of the students in this study. In nursing education, teaching and learning with real patients in a clinical setting is very much essential and it is indeed very difficult to manage with distance e-learning. The solution for this condition can be the use of virtual patients. Virtual patients are designed in such a way that it helps in simulating real scenario cases and they help in pre-preparing the students before facing new patients [21]. Technical, institutional and student related barriers was found to be the three main challenges in the implementation of e-learning. Lack of internet access, infrastructure and poor internet quality are the examples of such barriers that affect the e-learning [22].

## Effectiveness of e-learning against traditional face-to-face learning method

When assessing the effectiveness of e-learning over traditional learning, e-learning was found to be less effective for the respondents than traditional face-to-face learning. The finding is consistent with the findings of many other studies conducted in India, Nepal, and Pakistan [5, 13, 23, 24]. The probable reason could be that the students are more acquainted with the traditional learning activities and since nursing involves skill attainment tasks through practical, the objectives are not met through virtual learning. Another study conducted in Taiwan also indicated that face-to-face learning was perceived to be more effective than online learning in terms of all social presence, social interaction, and students' satisfaction [25]. A review conducted to assess the use of e-learning programs in nursing education found that students were more satisfied with lecture method than e-learning [26]. In contrast to the finding, a previous systematic review on e-learning has shown e-learning as equivalent to traditional learning in terms of academic context [20]. In one study conducted in India, students preferred to use a combination of face-to-face and e-learning education [13].

Classroom learning was given more preference by dental students rather than the distance e-learning in a study conducted among the undergraduate students in Indonesia because the latter method of learning resulted in problematic communication and gave less learning satisfaction [27]. However, one study found that students preferred the combination (hybrid) of face-to-face leaning and online learning [28].

### Attitude of respondents regarding e-learning

About half of the students in this study had positive perception of the usefulness of e-learning which is similar to the studies carried out in Nepal and Pakistan [10]. Students said that it saved their time and enabled them to improve their access to learning materials. Nevertheless, majority revealed that e-learning could never completely replace other forms of teaching and learning. Almost two-fifths of the students were neutral regarding the adaption of e-learning by the universities. In another study conducted to understand the challenges of e-learning almost half of the nursing students agreed to the importance of incorporating e-learning into the nursing curricula [18].

Regarding intention to adapt e-learning, all of the respondents wished for adapting e-learning in near future whereas, in a study conducted in Pakistan, only 85% wished to enroll and attend future e-learning courses [10]. About one-third of the students were planning to buy a computer to follow lecture notes easily while attending online classes. Contrary to this finding, another study reported only 23% of the respondents showing future e-learning preferences [5]. A study conducted during the pandemic to understand the perception of students' regarding online learning showed that majority of the participants did not want to continue this type of learning [27]. A study conducted in Jordan identified implementation of e-learning in medical education as challenging because though online learning could replace theoretical knowledge, replacing the clinical medical skills is not as easy as it appears to be. A combination of traditional and e-learning classes (hybrid) can be the desired way to deliver medical education in the future [22].

Regarding ease of learning, only half of the respondents had satisfactory responses. Though almost half of the respondents believed that it is almost easy, the other 54% and 52% felt that they are becoming a slave to technology and that the internet is making them slow correspondingly.

It is found that there is a lack of support from educational institutions to improve users' ability and skills in adapting e-learning which in turn has led the users to face difficulties in practicing e-learning. This has undoubtedly given rise to different reasons to avoid this type of learning. Intensive training programs are required to enhance user skills towards computer and e-learning technologies [6]. In this study, technical support was the major neutral point for the students with a mean score of only 2.9. Findings from other studies have also pointed out that insufficient technical support is a key challenge to fostering e-learning [29, 30]. Technical problems, such as an error in connection were the most important limitation to internet usage among Iran's medical students [31]. More than 85% of the respondents in this study were stressed with slow internet connections. Congruently, almost 30% felt anxious about their ability to use e-learning adding that to another level of stress. Taiwanese nurses in a study reported inconvenience experienced with the under-preparedness in terms of technical challenges posed by online learning and also expressed the frustration caused by technical problems [32]. Almost half of the respondents in this study were neutral regarding the provision of updated website in their colleges and whether the institutions are facilitating any e-learning training programs. A study conducted to identify the issues of adaption of e-learning in developing countries found high ICT (Information Communication and Technology) illiteracy

rates among the students and insufficient user/ technical support as the underlying causes of failure to adapt e-learning [29]. A study suggests that when students are comfortable using the internet, then it attracts them to prefer e-learning over traditional face-to-face learning [33].

Institutional support and institutional strategy plays a vital role in implementing the core skills and adaption of methodologies of e-learning [27]. In this study majority of the students remained neutral regarding the activities carried out by their institutions in relation to the technical aspects being provided by their institution for the delivery of e-learning.

Distance e-learning has been reported to provide more manageable and effective access [22]. The present study showed the respondents perceived distance learning as one of the effective was in the form of its ability to reach remote areas, reduction in the cost of travel, easy adaption by the married students to balance home and study (as there are females in majority of the nursing study) and as well as to being able to work and study from home. E-learning has provided a good alternative to the people located in a geographical location that is hard to reach and has difficult access to physical classes [34].

The mean scores for perceived usefulness, intention to adapt, and distant use of e-learning were more in comparison to ease of learning, technical support, and learning stressors in this study which is in line with the study conducted in Oman for the perceived usefulness, intention to adapt and distant use of e-learning however lower for the other domains: learning stressors, technical support and ease of learning [10].

## Overall attitude regarding e-learning

In a study conducted in Iran to assess the attitude of medical students, 43.4% considered e-learning useful for medical education [31]. In this study, almost 59% of the students had a favorable attitude toward e-learning. The result was similar to the other studies conducted in Nepal [24, 35]. Another study conducted in Pakistan also showed a favorable attitude of the students [5]. Whereas, in a study conducted in India, only 30.8% showed overall positive attitude [13]. A similar finding was found in another study as well [5]. Furthermore, the overall satisfaction rate in medical distance learning was only 26.8% in one of the studies conducted in Jordanian Universities, and it was significantly higher in students with previous experience in distance learning in their medical schools [22].

Several studies report the relationship of demographic variables with the attitude of the users of e-learning such as age, race and gender [36]. Whereas there was no association of attitude of the respondents regarding e-learning with the selected socio-demographic variable (age, residence, college, year of study, and ever being participated in e-learning before this pandemic) in this study. The finding is similar to the study conducted in West Bengal [37]. However, the finding was not supported by the finding of a study that revealed the association of e-learning with residence, and family income [25]. A study conducted in Jordan showed association of overall satisfaction of e-learning with previous experience in e-learning [38].

A study conducted to assess the acceptance of Internet-based learning medium, found that perceived usefulness and perceived enjoyment had direct effect on intention to use e-learning [39]. Similar to this finding, perceived usefulness had positive effect on intention to adapt in this study.

## Limitations of the study

The nature of the study is the main limitation as the result cannot be generalized. The study was conducted among nursing students of only four different colleges and may not be representative of the entire country.

## Conclusion

Although more than half of the respondents were positive regarding e-learning, students favor traditional face-to-face learning more. This can be due to practical issues and introduction of new phenomena of learning in our country. The nursing educational system should use the programs for improving e-learning that is more user-friendly and technically sound where virtual experiences of practical sessions can also be carried out effectively and efficiently. E-learning programs with proper strategies needs to be developed based on existing evidence to enhance the nursing students' clinical skills, knowledge and attitudes for the preparedness of the emergency like COVID-19 in the near future. The blended approach of teaching and learning in nursing fraternity can create new opportunities in nursing field in the coming days.

## Supporting information

**S1 Tool.**
(DOCX)

**S1 Dataset.**
(XLSX)

## Acknowledgments

The entire research team would like to thank all the participants of the study for providing their valuable time and information.

## Author Contributions

**Conceptualization:** Pratima Thapa.

**Data curation:** Pratima Thapa, Suman Lata Bhandari, Sharada Pathak.

**Formal analysis:** Pratima Thapa, Suman Lata Bhandari, Sharada Pathak.

**Investigation:** Pratima Thapa.

**Methodology:** Pratima Thapa.

**Project administration:** Pratima Thapa.

**Resources:** Pratima Thapa.

**Software:** Pratima Thapa.

**Supervision:** Pratima Thapa.

**Validation:** Pratima Thapa.

**Visualization:** Pratima Thapa.

**Writing – original draft:** Pratima Thapa.

**Writing – review & editing:** Pratima Thapa, Suman Lata Bhandari, Sharada Pathak.

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
