## [Decision Letter · Decision Letter 0]

18 Feb 2021

PONE-D-21-00551

Nursing students’ attitude on the practice of e-learning: A Cross-sectional survey amid Covid 19

PLOS ONE

Dear Dr. Thapa,

Thank you for submitting your manuscript to PLOS ONE. After careful consideration, we feel that it has merit but does not fully meet PLOS ONE’s publication criteria as it currently stands. Therefore, we invite you to submit a revised version of the manuscript that addresses the points raised during the review process.

We look forward to receiving your revised manuscript.

Kind regards,

Jenny Wilkinson, PhD

Academic Editor

PLOS ONE

Journal Requirements:

2. Thank you for submitting the above manuscript to PLOS ONE. During our internal evaluation of the manuscript, we found significant text overlap between your submission and the following previously published works, some of which you are an author.

Introduction section:

- https://www.researchgate.net/publication/342501833_The_impact_of_COVID-19_on_Education_in_Ghana

- https://www.jelsciences.com/articles/jbres1151.php

- http://www.pjms.org.pk/index.php/pjms/article/view/2766

Results & Discussion sections:

- https://www.semanticscholar.org/paper/Attitude-of-nursing-students-towards-e-learning-Ali-Jamil/6349db2f9f5bb9e1fcb06b7a84b423b47c713c13?p2df

- https://www.researchsquare.com/article/rs-41178/v1

- https://www.erudit.org/fr/revues/irrodl/2016-v17-n5-irrodl04876/1064708ar.pdf

- https://bmcmededuc.biomedcentral.com/articles/10.1186/s12909-020-02257-4

Please revise the manuscript to rephrase the duplicated text, cite your sources, and provide details as to how the current manuscript advances on previous work. Please note that further consideration is dependent on the submission of a manuscript that addresses these concerns about the overlap in text with published work.

3. Please include additional information regarding the survey or questionnaire used in the study and ensure that you have provided sufficient details that others could replicate the analyses.

For instance, if you developed the survey or questionnaire as part of this study and it is not under a copyright more restrictive than CC-BY, please include a copy, in both the original language and English, as Supporting Information.

If the questionnaire is published, please provide a citation to the (i) questionnaire and/or (ii) original publication associated with the questionnaire.

5. We note you have included ten tables to which you do not refer in the text of your manuscript. We also note that there are 8 table numbers in titles and table no.s 6 and 7 are duplicated in table titles.

Please ensure that you refer to all Tables in your text; if accepted, production will need this reference to link the reader to each Table.

Additional Editor Comments:

Thank you for your submission. Reviewer comments are provided for you and I now invite you to revise your work in response to these comments. One of the reviewers has noted that there is high similarity to published works and I ask that you ensure that appropriate attribution is used and that the work is in your own words. You may wish to seek additional help for English language to help improve the presentation of your work as some parts are awkwardly worded or have grammatical errors.

Reviewers' comments:

Reviewer's Responses to Questions

**Comments to the Author**

1. Is the manuscript technically sound, and do the data support the conclusions?

Reviewer #1: Yes

Reviewer #2: Partly

2. Has the statistical analysis been performed appropriately and rigorously? 

Reviewer #1: I Don't Know

Reviewer #2: No

3. Have the authors made all data underlying the findings in their manuscript fully available?

Reviewer #1: No

Reviewer #2: Yes

4. Is the manuscript presented in an intelligible fashion and written in standard English?

Reviewer #1: No

Reviewer #2: Yes

5. Review Comments to the Author

Reviewer #1: Dear Authors

thank you for the time you have taken for undertaking both this study and the committment to writing the manuscript. I think the paper could add to the body of literature around e-learning challenges during COVID however this study needs to be situated in the Nepalese context and the international literature. The methods section requires more work to meet the PLOS guidelines of "which focus on the technical aspects of a study rather than more subjective evaluations of issues like 'impact' or 'interest level' and I have made comments in the word document about this.Please review the attached documents that may assist you to strengthen the paper. I wish you well in publishing your work.

Reviewer #2: Although the manuscript describes a technically sound piece of scientific research with data which support the conclusions, the research methodology is too simplistic and the weak descriptive findings reported in the manuscript contribute little to existing knowledge. Many similar research studies (e.g. Regmi & Regmi, 2010) have been conducted.

Regmi, K. R., & Regmi, S. (2010). Medical and nursing students attitudes towards interprofessional education in Nepal. Journal of Interprofessional Care, 24(2), 150–167. https://doi.org/10.3109/13561820903362254.

A similarity test reported an index of 43%. The author is suggested to ensure this manuscript is free from any issues of plagiarism.

In addition to providing descriptive statistics and identifying statistical relationships between the five attitudinal dimensions and demographic variables, the author can consider exploring the statistical relationships among the five dimensions and overall attitudes. Statistical tests such as correlations, multiple regression, ANOVA and MANOVER can be considered.

Overall, the manuscript suffers from a lack of contribution of the research study reported and the simplistic research method.

6. PLOS authors have the option to publish the peer review history of their article (what does this mean?). If published, this will include your full peer review and any attached files.

Reviewer #1: No

Reviewer #2: No

---

## [Author Response · Author response to Decision Letter 0]

24 Mar 2021

The responses to the reviewers’ comments are set below 

Comments for authors

Resonse: 

We are very grateful for the academic editor’s suggestions. The manuscript has been modified as per the PLOS One’s style requirement, including for the file naming. The template has been followed.

2. Thank you for submitting the above manuscript to PLOS ONE. During our internal evaluation of the manuscript, we found significant text overlap between your submission and the following previously published works, some of which you are an author.

Introduction section:

- https://www.researchgate.net/publication/342501833_The_impact_of_COVID-19_on_Education_in_Ghana

- https://www.jelsciences.com/articles/jbres1151.php

- http://www.pjms.org.pk/index.php/pjms/article/view/2766

Results & Discussion sections:

- https://www.semanticscholar.org/paper/Attitude-of-nursing-students-towards-e-learning-Ali-Jamil/6349db2f9f5bb9e1fcb06b7a84b423b47c713c13?p2df

- https://www.researchsquare.com/article/rs-41178/v1

- https://www.erudit.org/fr/revues/irrodl/2016-v17-n5-irrodl04876/1064708ar.pdf

- https://bmcmededuc.biomedcentral.com/articles/10.1186/s12909-020-02257-4

Please revise the manuscript to rephrase the duplicated text, cite your sources, and provide details as to how the current manuscript advances on previous work. Please note that further consideration is dependent on the submission of a manuscript that addresses these concerns about the overlap in text with published work.

Response:

Manuscript has been revised and cited for the sources. Rationale for the conduction of the research has also been revised. 

3. Please include additional information regarding the survey or questionnaire used in the study and ensure that you have provided sufficient details that others could replicate the analyses.

For instance, if you developed the survey or questionnaire as part of this study and it is not under a copyright more restrictive than CC-BY, please include a copy, in both the original language and English, as Supporting Information.

If the questionnaire is published, please provide a citation to the (i) questionnaire and/or (ii) original publication associated with the questionnaire.

Response:

Additional information regarding the survey and questionnaire has been provided. 

The question has four parts. Part 1, 2 and 3 was developed by the authors themselves, and part 4 was obtained from previous study. Necessary permission was obtained from the author of the previous study to use the questionnaire in part 4 from the authors. Citation has been provided in the manuscript.

The questionnaire has been provided as a separate file as supporting information. 

Response:

Data file has been made available in the supporting information files. 

5. We note you have included ten tables to which you do not refer in the text of your manuscript. We also note that there are 8 table numbers in titles and table no.s 6 and 7 are duplicated in table titles.

Please ensure that you refer to all Tables in your text; if accepted, production will need this reference to link the reader to each Table.

Response:

The table number has been corrected and all tables has been referred in the text. 

Response:

Ethics statement from other section has been removed. 

Additional Editor Comments

Reviewers' comments:

Reviewer's Responses to Questions

Comments to the Author

1. Is the manuscript technically sound, and do the data support the conclusions?

Reviewer #1: Yes

Reviewer #2: Partly

 Manuscript has been modified as per the given suggestions and feedbacks. 

2. Has the statistical analysis been performed appropriately and rigorously?

Reviewer #1: I Don't Know

Reviewer #2: No

 Statistical portion has been changed. 

3. Have the authors made all data underlying the findings in their manuscript fully available?

Reviewer #1: No

Reviewer #2: Yes

 Data has been provided as supporting information files.

4. Is the manuscript presented in an intelligible fashion and written in standard English?

Reviewer #1: No

Reviewer #2: Yes

 Language has been corrected. 

5. Review Comments to the Author

Reviewer #1: Dear Authors

thank you for the time you have taken for undertaking both this study and the committment to writing the manuscript. I think the paper could add to the body of literature around e-learning challenges during COVID however this study needs to be situated in the Nepalese context and the international literature. The methods section requires more work to meet the PLOS guidelines of "which focus on the technical aspects of a study rather than more subjective evaluations of issues like 'impact' or 'interest level' and I have made comments in the word document about this. Please review the attached documents that may assist you to strengthen the paper. I wish you well in publishing your work.

Response: 

Thank you for your constructive feedback. The revision has been done as per the comments below. 

Page 2 – suggest using cessation or another word instead of shutting down, please edit throughout. 

Shutting down word has been edited with other words throughout the manuscript

Page 3 please remove etc from the paper entirely and list the extra problems

Etc has been removed 

Page 3 there appears to be different fonts used at times? 

Fonts has been made uniform. Font styles are used as per the guidelines of plos one.

Page 3 suggest editing this sentence please So, based on this the research team researcher was is interested in carrying out a study regarding the attitude of the nursing students’ attitudes towards e-learning

Edit has been done as per the suggestion.

Page 3 I understand English may not be the author’s first language and you are to be congratulated for writing this paper fully in English, that must be difficult at times. However the readership will expect correct English grammar and punctuation. Could you have someone go over the paper please and correct statements such as this was never been tried before resubmitting. I will not comment any further on grammar and punctuation however I have highlighted some in the attached PDF. 

Thank you for the feedback. I have tried my best to correct English grammar in the revised manuscript. 

Methods – please reference and provide more information on the study method chosen, the validity testing and reporting of the statistical analysis of the survey in some more depth. Using the equator website is recommended and the tools there will assist you to include relevant information in the paper. For cross-sectional studies you could use the STROBE tool. 

It has been revised.

Please remove all abbreviations such as B.Sc. 

Removed all abbreviations. 

Page 4 the phases of the study could be moved into a table. 

I don’t think Ethical consideration: should be included please write this in a full sentence. 

Ethical consideration has been written in full sentence.

The results section has a lot of tables and data reported is not discussed in the discussion section. I would reconsider the important or new or interesting results and present and report only on them. 

Important and new findings are discussed in the discussion. 

The discussion also has some paragraphs that would be better placed in the background section for example the two that begin In a study conducted in Iraq…… 

Background section has been provided with the data.

This paragraph is repetitive of the results section and should not be in the discussion The mean scores for perceived usefulness, intention to adopt, and distant use of e-learning were

3.06, 3.07, and 3.82 which is much more in comparison to ease and there are other paragraphs similar to this one. 

This has been removed.

The conclusion in the paper is very short and seems to generalise the paper in the international data. For example there are many nursing education providers that do and have shown effective, reliable and efficient use of e-learning and virtual learning experiences. I think overall a wider literature review and a more honed problem statements is needed for this paper to progress to publication. 

Conclusion has been re-written.

Also regarding the referencing the numbers need to be super script like so 1 

The references are made as per the feedback. 

There does appear to be some inconsistencies in the reference list such as some doi included and different presentations of the doi. Also see number 24 I am not aware of using cited as in the reference list, however I am not an expert on Vancouver style so I stand corrected if this is normal practice. 

Vancouver style referencing list has been corrected. 

Reviewer #2: Although the manuscript describes a technically sound piece of scientific research with data which support the conclusions, the research methodology is too simplistic and the weak descriptive findings reported in the manuscript contribute little to existing knowledge. Many similar research studies (e.g. Regmi & Regmi, 2010) have been conducted.

Regmi, K. R., & Regmi, S. (2010). Medical and nursing students attitudes towards interprofessional education in Nepal. Journal of Interprofessional Care, 24(2), 150–167. https://doi.org/10.3109/13561820903362254.

A similarity test reported an index of 43%. The author is suggested to ensure this manuscript is free from any issues of plagiarism.

In addition to providing descriptive statistics and identifying statistical relationships between the five attitudinal dimensions and demographic variables, the author can consider exploring the statistical relationships among the five dimensions and overall attitudes. Statistical tests such as correlations, multiple regression, ANOVA and MANOVER can be considered.

Overall, the manuscript suffers from a lack of contribution of the research study reported and the simplistic research method.

 Correlation has been established between the domains and also with the overall attitude regarding e-learning. 

6. PLOS authors have the option to publish the peer review history of their article (what does this mean?). If published, this will include your full peer review and any attached files.

Do you want your identity to be public for this peer review? For information about this choice, including consent withdrawal, please see our Privacy Policy.

Reviewer #1: No

Reviewer #2: No

Yes, the figures have been done as per the guidelines.

---

## [Editor Report · Decision Letter 1]

30 Mar 2021

PONE-D-21-00551R1

Nursing students’ attitude on the practice of e-learning: A Cross-sectional survey amid Covid 19 in Nepal

PLOS ONE

Dear Dr. Thapa,

Thank you for submitting your manuscript to PLOS ONE. After careful consideration, we feel that it has merit but does not fully meet PLOS ONE’s publication criteria as it currently stands. Therefore, we invite you to submit a revised version of the manuscript that addresses the points raised during the review process.

We look forward to receiving your revised manuscript.

Kind regards,

Jenny Wilkinson, PhD

Academic Editor

PLOS ONE

Additional Editor Comments (if provided):

Thank you for the revisions and responses to reviewer comments.

1. There are a number of grammatical and other language issues in the work and I strongly recommend that the work is reviewed by a native English writer or by a professional editing service.

2. In the new text describing the sample size from 4 institutions is stated as 482; this seems rather small. Please confirm that this is the total number of students studying nursing at these 4 institutions; if not then please provide clarification what the number refers to.

3. In this same section (sample size) the distribution of survey invitations via mail (should this be e-mail), WhatsApp etc is stated however it is unclear how the authors determined the overall sample size to calculate response rate or what database was used to generate the invitations to participate.

4. Please provide details of the pre-testing of the instrument

5. Numerical data should be given to a consistent number of decimal places, for example in the first paragraph of the section ‘Sociodemographic data’ some data is given to 1 decimal place while others (for the same parameter) is given to 2 decimal places.

6. For in text citation do not include the author initial

---

## [Author Response · Author response to Decision Letter 1]

14 Apr 2021

Additional Editor Comments (if provided):

Thank you for the revisions and responses to reviewer comments.

1. There are a number of grammatical and other language issues in the work and I strongly recommend that the work is reviewed by a native English writer or by a professional editing service.

Response: The work has been reviewed by an English teacher. 

2. In the new text describing the sample size from 4 institutions is stated as 482; this seems rather small. Please confirm that this is the total number of students studying nursing at these 4 institutions; if not then please provide clarification what the number refers to.

Response: Yes, the total number of the students in 4 selected colleges was 482. 

3. In this same section (sample size) the distribution of survey invitations via mail (should this be e-mail), WhatsApp etc is stated however it is unclear how the authors determined the overall sample size to calculate response rate or what database was used to generate the invitations to participate.

Response:Total enumerative sampling method was used. So, all the students were enrolled in the study with sample size of 482. 

4. Please provide details of the pre-testing of the instrument

Response: The validity of the instrument was maintained by extensive literature review and consultation from subject experts. The instrument was translated to native language and then again translated to English version. The reliability of the instrument was examined for internal consistency by pre-testing the instrument in 10% (48 nursing students) of a similar type of estimated population in a similar setting. Necessary modification in the questionnaire was done as per the results obtained. The reliability score for the instrument for part 2 and 3 was found to be 0.98 on pretesting. 

5. Numerical data should be given to a consistent number of decimal places, for example in the first paragraph of the section ‘Sociodemographic data’ some data is given to 1 decimal place while others (for the same parameter) is given to 2 decimal places.

Response:All numerical data has consistent number of decimal after correction. 

6. For in text citation do not include the author initial

Response: Author initials has been removed from the text citation.

As per the suggestion the figure from the manuscript file has been removed.

---

## [Editor Report · Decision Letter 2]

26 Apr 2021

PONE-D-21-00551R2

Nursing students’ attitude on the practice of e-learning: A Cross-sectional survey amid COVID-19 in Nepal

PLOS ONE

Dear Dr. Thapa,

Thank you for submitting your manuscript to PLOS ONE. After careful consideration, we feel that it has merit but does not fully meet PLOS ONE’s publication criteria as it currently stands. Therefore, we invite you to submit a revised version of the manuscript that addresses the points raised during the review process.

We look forward to receiving your revised manuscript.

Kind regards,

Jenny Wilkinson, PhD

Academic Editor

PLOS ONE

Journal Requirements:

Additional Editor Comments (if provided):

Thank you for your revisions, unfortunately there are still a number of language and grammatical issues that need attention. I strongly recommend use of either a professional editing service or someone who has extensive experience in writing for publication.

---

## [Author Response · Author response to Decision Letter 2]

5 May 2021

Response: Referencing has been corrected. There are no any references from retracted papers.

Thank you for your revisions, unfortunately there are still a number of language and grammatical issues that need attention. I strongly recommend use of either a professional editing service or someone who has extensive experience in writing for publication.

Response: Professional editing service was used in editing language and grammatical issues.

---

## [Editor Report · Decision Letter 3]

10 Jun 2021

Nursing students’ attitude on the practice of e-learning: A Cross-sectional survey amid COVID-19 in Nepal

PONE-D-21-00551R3

Dear Dr. Thapa,

We’re pleased to inform you that your manuscript has been judged scientifically suitable for publication and will be formally accepted for publication once it meets all outstanding technical requirements.

Kind regards,

Jenny Wilkinson, PhD

Academic Editor

PLOS ONE
---

## [Editor Report · Acceptance letter]

15 Jun 2021

PONE-D-21-00551R3 

Nursing students’ attitude on the practice of e-learning: A Cross-sectional survey amid COVID-19 in Nepal. 

Dear Dr. Thapa:

I'm pleased to inform you that your manuscript has been deemed suitable for publication in PLOS ONE. Congratulations! Your manuscript is now with our production department. 

Kind regards, 

on behalf of

Dr Jenny Wilkinson 

Academic Editor

PLOS ONE